# Sparse representation learning derives biological features with explicit gene weights from the Allen Mouse Brain Atlas

**Mohammad Abbasi**[1], **Connor R. Sanderford**[1], **Narendiran Raghu**[1], **Mirjeta Pasha**[2], **Benjamin B. Bartelle**[1] *

**1** School for Biological and Health Systems Engineering, Arizona State University, Tempe, Arizona, United States of America, **2** Department of Mathematics, Tufts University, Medford, Massachusetts, United States of America

* Benjamin.Bartelle@asu.edu

**Data Availability Statement:** All data used is currently public. The manuscript contains links to our Github with all analysis annotated with the specific datasets used.

## Abstract

Unsupervised learning methods are commonly used to detect features within transcriptomic data and ultimately derive meaningful representations of biology. Contributions of individual genes to any feature however becomes convolved with each learning step, requiring follow up analysis and validation to understand what biology might be represented by a cluster on a low dimensional plot. We sought learning methods that could preserve the gene information of detected features, using the spatial transcriptomic data and anatomical labels of the Allen Mouse Brain Atlas as a test dataset with verifiable ground truth. We established metrics for accurate representation of molecular anatomy to find sparse learning approaches were uniquely capable of generating anatomical representations and gene weights in a single learning step. Fit to labeled anatomy was highly correlated with intrinsic properties of the data, offering a means to optimize parameters without established ground truth. Once representations were derived, complementary gene lists could be further compressed to generate a low complexity dataset, or to probe for individual features with >95% accuracy. We demonstrate the utility of sparse learning as a means to derive biologically meaningful representations from transcriptomic data and reduce the complexity of large datasets while preserving intelligible gene information throughout the analysis.

## Introduction

Dimensionality reduction, manifold learning, and clustering are essential methods for processing transcriptomic data into "representations" of biology interpretable to the human mind [1]. A typical workflow can include Principle Component Analysis (PCA) [2], graph embedding with t-Distributed Stochastic Neighborhood Embedding (t-SNE) [3] or Uniform Manifold Approximation and Projection (UMAP) [4], and separation by Leiden clustering [5]. This approach is robust, but presents several weaknesses: Initial dimensionality reduction filters for high variance global trends over localized features, offering low sensitivity to rare or low expressing genes [6]. Manifold Learning methods, UMAP and t-SNE, conversely, are more sensitive to local structures of the data, but do not report the weights of input genes [7].

**Funding:** The authors have received no specific funding for this work.

**Competing interests:** The authors have declared that no competing interests exist.

**Abbreviations:** AMBA, Allen Mouse Brain Atlas; AMI, Adjusted Mutual Information; ARA, Allen Reference Atlas; ARI, Adjusted Rand Index; CA, Hippocampal Cornu Ammonis; CP, Caudate Putamen; CUL4,5, Lobules IV-V; DG, Dentate Gyrus; DLSC, Dictionary Learning and Sparse Components; GO, Gene Ontology; ICA, Independent Component Analysis; ISH, In Situ Hybridization; KPCA, Kernel Principal Component Analysis; L-BFGS, Limited-memory Broyden, Fletcher, Goldfarb, and Shanno; MERFISH, Multiplexed Error-Robust Fluorescence In Situ Hybridization; MI, Mutual Information; MOB, Main Olfactory Bulb; PCA, Principal Component Analysis; ROC-AUC, Receiver Operating Characteristic-Area Under the Curve; SFt, Sparse Filtering; SPCA, Sparse Principal Component Analysis; SVD, Singular Value Decomposition; t-SNE, t-Distributed Stochastic Neighborhood Embedding; UMAP, Uniform Manifold Approximation and Projection; WT, Wild Type.

Clustering convolves noise within informative groupings [8]. Overall, each learning step further abstracts representation from data such that the contribution of any one gene to a cluster is not explicit.

Well described and unique markers solve most of these issues, offering the essential ground truth to connect points on a graph to biological states and a starting point for Gene Ontology (GO) analysis with experimental validation. Many representations lack a unique marker however and can only be described by the relative expression levels of common genes. Transcriptomic data lacking cellular resolution cannot be fully described by a singular marker even if one were present [9]. In these cases, each step of an analysis pipeline further obscures what a cluster might represent, requiring supervised learning approaches and careful validation to parse gene information.

Given the incredible advances in applied information theory [10], we asked if an unsupervised learning method could derive localized features within a dataset, while retaining actionable information about the inputs. Essentially, we asked if it was possible to generate representations of biology from transcriptomic data, in a single step, without implicit priors. For a more advanced approach to dimensionality reduction and clustering, we explored "sparse" learning methods, optimized for building localized features from a minimal number of input elements sparsely distributed within a large dataset [11].

To test if derived features were representations of biology and not just the isolation of a unique marker gene, we used a dataset with well described biological phenomena as a testable ground truth. The Allen Mouse Brain Atlas (AMBA) is the most comprehensive spatial transcriptomic dataset of the whole brain available during this analysis. What makes AMBA the optimal test data for our methods is that every voxel of transcriptomic data has been registered to the Common Coordinate Framework (CCF) and labeled as an element of anatomy based on thousands of person-hours worth of labor [12]. The low resolution (200μm isotropic) of AMBA convolves multiple cell types, making single gene markers uncommon and an incomplete descriptor of any element of neuroanatomy [13]. While the developmental genes involved in forming different brain structures are mostly inactive in the adult mouse brain, gene expression can still functionally define anatomy [13]. These data therefore offer a means to test the ability of a learning method to describe established elements of anatomy with derived signatures of gene expression [13, 14].

We surveyed previously established methods and new advances to find that learning methods with sparsity constraints offered the unique potential to return biologically meaningful representations without the need for secondary clustering steps. With some parameter tuning, all sparse representation learning methods tested for dimensionality reduction could generate anatomical features with a corresponding ranked list of genes. Moreover, gene lists could be compressed to a minimal ensemble of markers for each element of anatomy while retaining high fidelity to ground truth. Overall, sparse learning methods offer a means to derive biologically representative features and descriptive minimal gene lists from transcriptomic data.

## Results

To survey representation learning methods and match previous analyses, we first established metrics for comparing representations derived from the high-fidelity coronal In Situ Hybridization (ISH) data for 2,941 transcripts from AMBA with ground truth anatomical information of CCF, to test the principles behind the algorithms that generated them [12]. Because anatomy is presented as labels, we initially applied secondary clustering using K-means for all methods so that our results were directly comparable to previously published studies [13, 15]. All analyses were done using the whole brain volume with 100 representations derived using

all dimensionality reduction methods surveyed here to match previous studies, then compared against CCF volumetrically. CCF labels 574 unique brain structures at 200 μm isotropic resolution. Because many smaller brain regions are only a few voxels large, even if every anatomical region was perfectly defined by transcription, biological variance, resolution, and overfitting artifacts introduce significant error that was unavoidable using unsupervised methods, but common to all approaches.

## Mutual information based feature learning with clustering derives low resolution representations of anatomy from the AMBA dataset

First postulated in information theory [16], the maximum information principle presents a model of information transfer that has since been used to filter complex data [17]. Infomax based algorithms build representations by separating low Mutual Information (MI), or independent, components of a set, and grouping high MI, or the most mutually interdependent ones. InfoMax is central to independent component analysis (ICA) and a commonly-used objective in many other machine learning methods, such as variational autoencoders [18, 19]. ICA was recently used on similarly low resolution brain-wide spatial transcriptomic data, so we asked if ICA components would recapitulate anatomy in AMBA in a similar manner to sequencing based approaches [14].

ICA generated components that were broadly active across the brain, but with areas of high intensity that appeared anatomical (**Fig 1A**). To filter for highly active areas and parcelate the data into labels, we applied K-means clustering, resulting in representations that were more visually similar to anatomy, though less detailed (**Fig 1B**). To quantify how well clustered

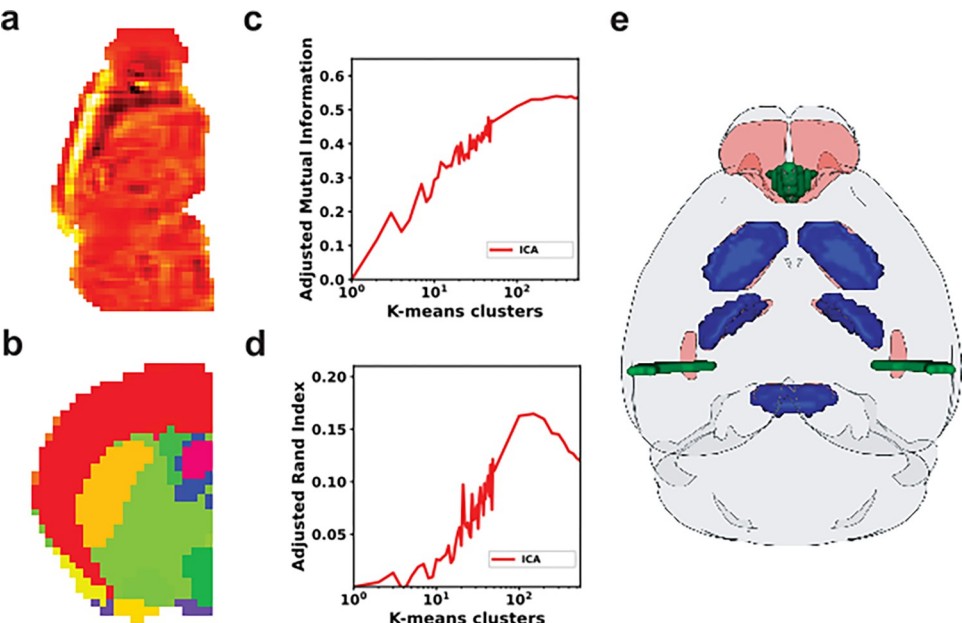

**Fig 1. Representation learning on AMBA with ICA.** In a whole brain analysis of 100 ICA derived representations **(a)** A selected representation shows activity across the brain with a bright anatomical like region. **(b)** K-means clustering of the ICA representations appears more selectively anatomical (K = 200). **(c)** AMI and **(d)** ARI of clustered components, compared to neuroanatomy, using K-means clustering with K ranging from 1 to 50 and from 50 to 550 with a step of 50. **(e)** 3D representation of K-means clusters with selected top overlapping clusters (in blue) with brain regions (Olfactory Tubercle, Reticular nucleus of the Thalamus, and Pontine Gray) in red and low overlapping clusters (in green) with brain regions (Basolateral Amygdalar Nucleus, posterior part and Main Olfactory Bulb) in red using Dice similarity coefficient.

representations fit anatomy, we used Adjusted Mutual Information (AMI) and Adjusted Rand Index (ARI) to test if a region labeled by clustering was co-labeled by anatomy, then generated scores from 1 to 550 k-means clusters to look for an optimal fit to ground truth. Peak fitting occurred at 300 clusters by AMI (**Fig 1C**) and 150 by ARI (**Fig 1D**), well below the 574 labeled elements of anatomy in CCF, but more than the 181 clusters derived using ICA/K-means on a less comprehensive whole brain transcriptome [14].

Visual inspection of clustered representations showed the poorest fits to anatomy, by Sørensen–Dice similarity coefficient, with poor representations appearing at interfaces like olfactory bulb and nucleus, or within a single coronal slice, which was the orientation of the high-fidelity In Situ Hybridization (ISH) experiments of AMBA used in this analysis (**Fig 1E**). Similar artifacts were previously reported with ICA of spatial transcriptomic data, requiring manual curation before subsequent clustering and parcellation by supervised learning methods [14]. Without applying these additional steps, ICA/K-means derived representations offered the lowest similarity to AMBA labels of any method tested.

## Variance based feature learning with clustering generates improved representations of anatomy, with no advantage to nonlinear models

The earliest analysis of the AMBA data used Singular Value Decomposition (SVD) and K-means clustering to generate representations that resembled anatomy [13]. SVD is the basis for PCA, which similarly projects the data to a subspace of maximal variance for evaluation of a subset of components. We applied and chose 100 representations for uniform comparison with previous methods. PCA of AMBA generated components with high intensity areas that visually recapitulate anatomy, but are active across most of the brain, similar to ICA (**Fig 2A**). K-means clustering on the full ensemble of PCA components parcellated brain voxels into finer and more anatomical looking representations of anatomy than ICA (**Fig 2B**).

Biological relationships are not always linear, so we asked if a nonlinear approach could more accurately reflect underlying biology. Unlike the eigendecomposition of PCA, Kernel PCA (KPCA) components are fit to a preselected kernel function. Parameter tuning across multiple nonlinear transformations however, did not produce visually different components from those generated by PCA (**Fig 2C**). Applying K-means to KPCA components produced similar parcellations to PCA/K-means (**Fig 2D**). Quantitative comparison of linear and nonlinear methods demonstrated no advantage to KPCA, with some kernels performing worse at every value of k, suggesting no advantage to non-linear methods.

## Sparsity constrained feature learning with clustering generates the most accurate anatomical representations

Sparse representation learning algorithms generally satisfy infomax principles but apply sparsity constraints to one or more aspects of their outputs: the number of representations that are informative of the input data (feature or population sparsity), the samples of data that comprise any given representation (sample or lifetime sparsity), and the distribution of informative samples across all representations (dispersal) [20]. For DLSC and SPCA, the sparsity variable $\alpha$ is a regularization parameter, which imposes sparsity on components in DLSC and loadings in SPCA. SFt does not have a sparsity parameter, and instead uses a nonlinear transformation, ridge ($l_2$) regression for normalization across features, a second $l_2$ normalization across samples and the least absolute shrinkage and selection operator ($l_1$) regularization for global minimization [20, 21]. Ideally sparse representations would then be non-overlapping, with each one presenting similar levels of informativeness regarding the input data. Previous work has

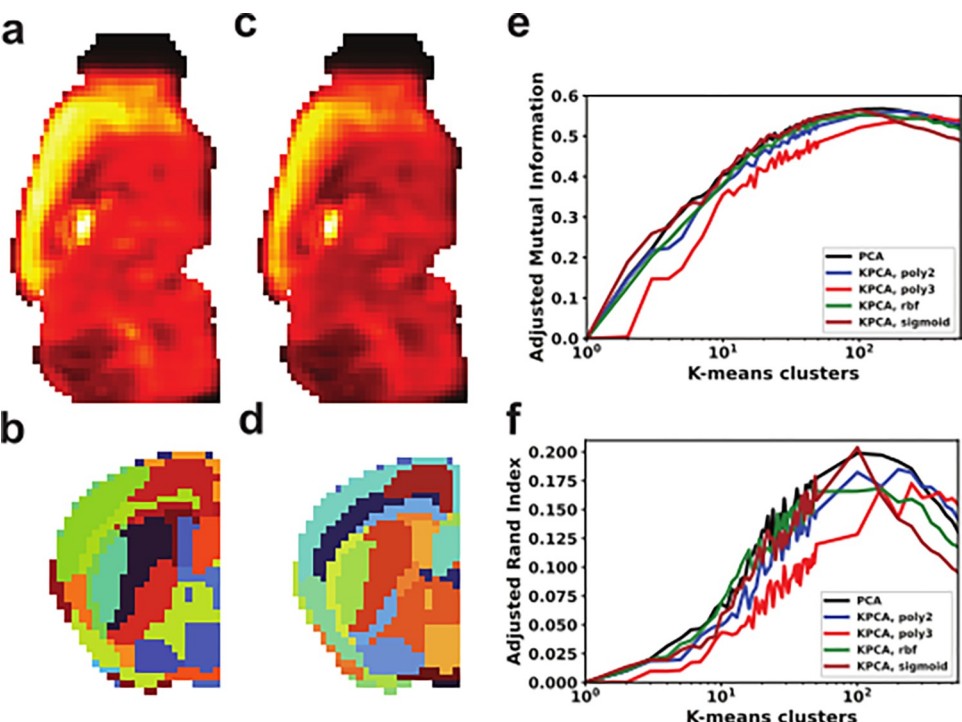

**Fig 2. Linear and nonlinear PCA generate equivalent representations of anatomy from AMBA. (a)** First PCA component representation in a sagittal slice. **(b)** Representation of K-means clustering (K = 200) of PCA components in a coronal slice of the left hemisphere. **(c)** First KPCA component with Quadratic kernel function in a sagittal slice. **(d)** Representation of K-means clustering (K = 200) of KPCA components with Quadratic kernel function in a coronal slice of the left hemisphere. **(e, f)** Adjusted Mutual Information and Adjusted Rand Index of clustered components using K-means clustering with K ranging from 1 to 50 and from 50 to 550 with a step of 50.

claimed AMBA representations generated with Dictionary Learning and Sparse Components (DLSC), outperformed PCA when combined with clustering [15].

We generated representations starting with previously described optimal hyperparameters, including a low sparsity hyperparameter ($\alpha = 0.1$) [15]. Sparsely derived features covered similar anatomy as other tested methods with similarly broad activity (**Fig 3A, 3C and 3E**). As previously reported, clustering of DLSC resulted in similar but visually finer representations than those generated by PCA (**Fig 3B**). Sparse PCA (SPCA) performed similarly with the same hyperparameters, generating larger representations with comparable anatomical features to DLSC (**Fig 3B and 3C**) We then tried Sparse Filtering (SFt), a method that derives its own sparsity levels. SFt generated uniquely compact features, with activation in single anatomical-like regions (**Fig 3E**). SFt + K-means clustering yielded fine features, including the clearest depiction of cortical layers of any method tested (**Fig 3F**).

We quantified similarity to labeled anatomy across all sparse methods with fixed parameters across a range of clusters, showing greater accuracy for SFt than any other method, with a peak score of 0.61 at 200 clusters based on AMI scores (**Fig 3G**). ARI scores showed SFt performing substantially better, peaking at 0.28 with 50 clusters. In all methods, AMI scores peaked well below the 574 labeled brain regions, with most fitting optimally at ~200 clusters. The most accurate representations were predominantly subcortical, with resolution preventing an even more optimal fit to anatomy. Some visually obvious artifacts appeared as single slices or matched artifacts in the original histology data (**Fig 1G**). Visual inspection of each feature suggested improved fits were in part due to fewer coronal plane artifacts, but there is no way to

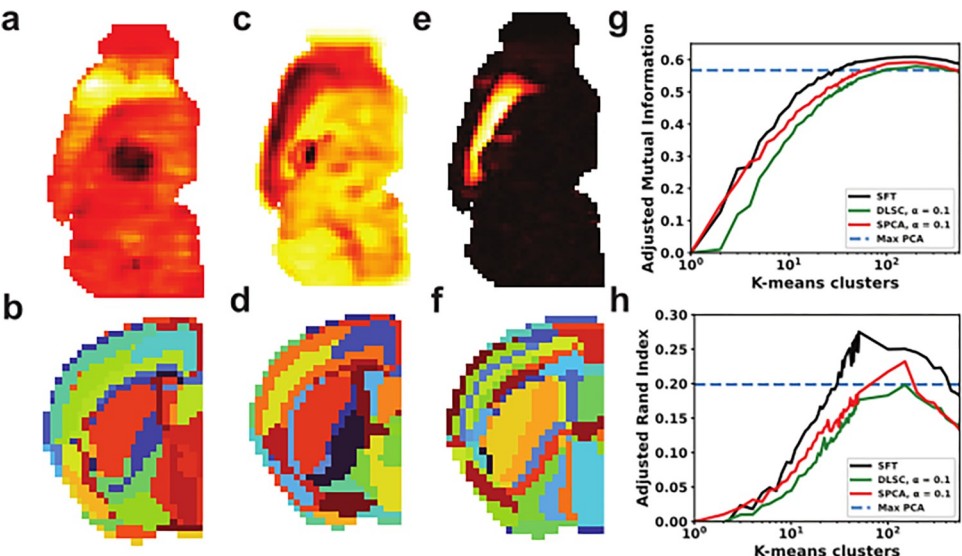

**Fig 3. Sparse representations of AMBA. (a)** DLSC representation, active in the cortex. **(b)** K-means clustering (K = 200) of DLSC representations in a coronal slice. **(c)** SPCA representation, chosen for similarity. **(d)** K-means clustering of SPCA representations. **(e)** SFt representation, active in the cortex. **(f)** K-means clustering of SFt components. **(g, h)** Adjusted Mutual Information and Adjusted Rand Index of clustered components using K-means clustering with K ranging from 1 to 50 and from 50 to 550 with a step of 50.

determine if the remaining mismatch between SFt and AMBA labels are due to the limits of resolution.

## Sparse learning methods without clustering can represent anatomy when optimized for minimal, spatially contiguous features

The uniquely sparse and compact features from SFt suggested no secondary clustering was required to describe anatomy and we asked if this could be achieved with other sparse methods after sparsity parameter tuning using $\alpha$ values from {0.1, 1, 10, 20}. To optimize DLSC and SPCA for better representations, we established descriptive metrics of spatial properties, similarity to ground truth, and underlying gene information: Shannon entropy measures the total information contained in all features. Sørensen–Dice similarity coefficient is a measure of overlap between a representation and the best fitting anatomical ground truth. The number of connected components describes the contiguity of a representation. Feature sparsity measures the exclusivity of activation to a given representation, with a single activated point being maximally sparse. Spatial entropy measures the total homogeneity of representation space. Finally, weight sparsity in the samples comprising each representation.

Across all conditions, Shannon entropy remained the same, indicating that no information was lost using a given method, only distributed differently depending on the method and parameters (**Fig 4A–4C**). As a baseline, metrics were applied to the expression of individual genes (**Fig 4A**). Without a clustering step PCA made poor representations of anatomy. For individual PCA components, spatial entropy remained high, with a lower connected components score and poor fits to anatomy by Dice coefficient (**Fig 4B**).

SFt offered the highest Dice coefficient of any method, even after parameter tuning, demonstrating that individual representations fit directly with ground truth anatomy (**Fig 4B**). This correlated with a high score for connected components, ideally one anatomical region per representation. A high feature sparsity score indicated spatially regions of high activation,

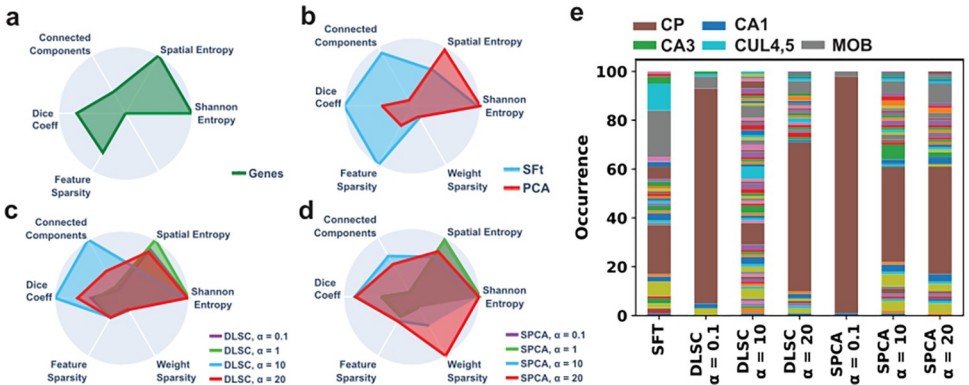

**Fig 4. Relative performance metrics of DLSC, SPCA, and SFt.** Representations derived from each method without additional clustering were evaluated along 5 metrics with increasing sparsity values ($\alpha$) when possible. Metrics, minus weight sparsity, applied to individual genes **(a)** PCA had the poorest metrics, while SFt had the best Dice coefficients of any method, correlating with feature sparsity and connected component score **(b)**. With increasing $\alpha$, DLSC representations lose spatial entropy and gain feature sparsity, with fewer connected components and improvements in individual fit to ground truth as measured by Dice coefficient **(c)**. For SPCA, higher $\alpha$ increases weight sparsity while Dice and connected components peak at $\alpha = 10$ **(d)**. Labeling all representations by the anatomy they best represent showed some brain regions were redundantly represented by all methods, with the caudate putamen (CP) overrepresented across all methods and $\alpha$ values, with representation diversity peaking with Dice **(e)**.

while spatial entropy score further described low activity across most of anatomical space. A similar weight sparsity score to PCA indicated that a similar number of genes were used to generate each representation. Overall SFt derived sparse features from the data, but did not derive sparse weightings to generate those features.

For DLSC, previous work reported a peak AMI score with clustering when the sparsity parameter $\alpha$ was low. Increasing $\alpha$ however improved Dice coefficient for direct representations, peaking at 100X the previously reported optimal value for AMI (**Fig 4C**). Peak Dice coefficient again correlated with peak connected component score, and comparable spatial entropy to SFt, indicating that information consolidated within features with accuracy until becoming incoherently sparse. Feature sparsity did not change with parameter tuning however, nor did weight sparsity, suggesting that the contents of DLSC features did not change substantially with tuning.

For SPCA, the average Dice coefficient remained lower than DLSC or SFt with optimal parameter tuning (**Fig 4D**). Like DLSC and SFt the Dice score, correlated with the number of connected components, with no relationship to spatial entropy or feature sparsity. At high $\alpha$, optimization was limited to weight sparsity, meaning the number of genes used to build each representation compressed without changing representation accuracy.

Comparing each feature across all methods and conditions, Dice coefficients were highly correlated with the connected components score ($r = 0.97$, $p < 0.01$). Individual measures of sparsity across features and weights did not reflect general properties of a biologically relevant feature.

## Sparse learning without clustering more optimally represents anatomy, when the number of redundant features is minimized

Dice coefficient and connected component scores described how well a representation fit a single element of anatomy, however upon labeling which elements of anatomy returned an optimal Dice score for each of the 100 features generated, we found some brain regions were

redundantly represented. For SFt, half of the features represented just 3 regions, for a total of 33 unique representations.

In DLSC and SPCA overlap was more extreme at the lowest α tested, with ~95% of features being representations of the CP (**Fig 4E**). Secondary clustering with K-means consolidated this overwhelming association into one label, with the remaining information describing the rest of anatomy and surprisingly giving an optimal fit to ground truth by AMI score [15]. Increasing α improved distribution of informativeness, generating more diverse features while improving the Dice coefficient. This was optimal in DLSC, with 64 unique representations at the same α that returned optimal Dice coefficient. Overshooting optimal sparsity caused a collapse back to redundant representations with an overall poorer fit. For SPCA, just as increasing α beyond minimal levels did not improve the Dice coefficient, the number of unique representations did not increase. Across all methods and conditions, more unique features correlated with the Dice coefficient (r = 0.91, p<0.01).

## Sparse learning derived features retain intelligible information as a weighted list of genes

While information is distributed across every feature derived from the data, the genes that comprise each one have explicit weightings. Using InfoMax and variance-based methods alone, weighting is spread across much of the data, but added sparsity constraints, can return a gene list that is itself sparse, with fewer genes of greater weight (**Fig 4A–4D**). SPCA, with hyperparameter tuning, returned maximal sparsity of gene weights, however this did not correlate with optimal representation accuracy by Dice coefficient (**Fig 4D**). Just as secondary clustering can filter less relevant information, we asked if the gene weights from one step representation learning could be compressed to the most relevant markers, while retaining representation fidelity. We chose SFt representations for their higher accuracy by Dice coefficient to see the effects of reducing the number of input genes (**Fig 4B and 4D**).

We increasingly compressed the AMBA dataset by generating SFt representations, establishing a weighted gene list for each representation, eliminating low ranked genes from the weighted gene lists, then repeating SFt on the new data subset until only the single highest ranked gene for each original representation was left. Representation fidelity degraded linearly with elimination of low rank genes when compared against SFt labels from a full dataset (**Fig 5A**), however compared to anatomical ground truth, representations from the compressed data did not see a performance drop until the gene samples fell were reduced to under 1000 genes then fell with a shallower slope, retaining 95% fidelity to anatomy at over 80% compression, or 584 genes (**Fig 5B**). Inaccuracies at this stage were limited to the edges of anatomical regions, suggesting the initial loss of fidelity was masked by resolution and systemic variance (**Fig 5C**).

We then asked if the gene information from a single SFt representation could be compressed using a supervised approach. We took the highest fidelity SFt representations by Dice coefficient, hippocampal cornu ammonis (CA) 1, CA3, and the CP, as labels, then used the top 10 most weighted genes from each to train a logistic regression-based classifier (**Fig 5D**). Of the most highly weighted genes, SFt picked several known markers for each region of the hippocampus CA1 [22], CA2 [23], CA3 [22], and Dentate Gyrus (DG) [24] (**Table 1**). CP has high molecular diversity and uniquely expressed genes, several of which were selected by SFt. Highly weighted genes included known markers of each anatomical region, but also genes expressed across multiple elements of anatomy or even markers of adjacent anatomy that could define a border (**Table 1**). Data was randomized over 5 iterations, taking 80% for training and the remaining 20% and accuracy measured by Receiver Operating Characteristic

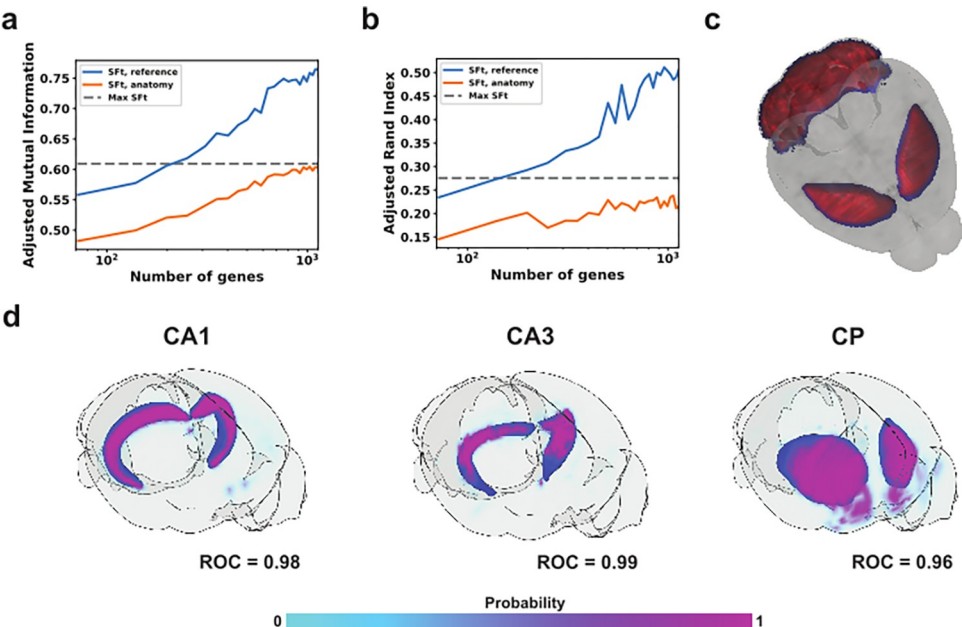

**Fig 5. Performance metrics of compressed gene sets.** Lower ranked genes across all SFt representations were iteratively removed from AMBA data, followed by SFt/clustering on the compressed dataset. **(a)** Adjusted Mutual Information between the full dataset vs. compressed representations and anatomy vs compressed representations. **(b)** Adjusted Rand Index of compressed representations vs full representations and anatomy. **(c)** Visualization of a selected feature derived from the SFt of the 2941-gene dataset (in red) with highest Pearson correlation to the 584-gene dataset (in blue), where 95% of the AMI score with reference is met. **(d)** Ten genes with the largest associated weights were used to train classifiers for their corresponding brain region. Probability of positive classification is shown by the colorbar, and the corresponding anatomy shown in blue. ROC score for five-fold cross-validation is shown for each method.

(ROC) Precision-Recall (PR) and curves. Classification was highly accurate with AUROC scores above 95% for every permutation. In order to account for the class imbalance, we also measured the AUPRC scores for CA1 (0.80), CA3 (0.78), and CP (0.74). The scores are high compared to base chance of classifying voxels in the representing feature for CA1 (0.02), CA3 (0.01), and CP (0.05). This demonstrated that an additional supervised learning step on a

**Table 1. 12 genes with highest weights for the overlapping SFt representation with the brain regions, with selectively expressed genes of a given brain region in bold.**

| Brain regions | CA1 | CA2 | CA3 | DG | CP |
|---|---|---|---|---|---|
| Genes | Pkp2 | Fibcd1 | Cyp7b1 | Crlf1 | Dock10 |
| | Tgfb2 | Arl15 | **Bok** | **Prox1** | Serpina9 |
| | **Tdo2** | C630041L24Rik | Cpne8 | Tdo2 | Prox1 |
| | Prox1 | Sstr4 | Tnfaip8l3 | Tnip2 | Rasd2 |
| | Camk1d | TC1412430 | Tanc1 | Dock10 | Tdo2 |
| | Crlf1 | Matn2 | Lats2 | C78409 | Cyp7b1 |
| | Il1rap | Cpne8 | 9930021J17Rik | Il1rap | C78409 |
| | Lpl | **Arhgap12** | Fibcd1 | C1ql2 | Rasgrp2 |
| | Dock10 | **Dock10** | Mycl1 | Cyp7b1 | Crlf1 |
| | Sms | Kcnd2 | Lhfpl2 | Pip5k1a | Pip5k1a |
| | Lct | Sh3rf1 | Iyd | Lct | C630035N08Rik |
| | C230081A13Rik | Prox1 | Camk1d | Syt6 | Dach1 |

representation weight matrix can compress the list of genes that describe it with nominal loss of representation fidelity.

## Discussion

In a broad comparison of representation learning methods, sparse algorithms generated more accurately described anatomical ground truth. SFt in particular generated direct anatomical representations without the need for clustering. More comprehensive metrics demonstrated that, with hyperparameter tuning, DLSC and SPCA could produce nearly comparable representations of anatomy. Each method optimized for different aspects of sparsity in the dataset with SFT deriving the highest feature sparsity, SPCA deriving more weight sparsity, and DLSC distributing informativeness across the most features. Sparse derived features without clustering presented a weighted gene list that could be compressed down to the most relevant genes for analysis, offering a minimal probe list for deriving specific elements of anatomy.

It is important to note that none of the learning methods tested considered spatial aspects of the data, meaning that adjacent voxels would only be included in a representation if they shared a molecular feature. This is the only basis for comparing learning methods and suggests that transcriptional signatures are sparse, or only found in the few voxels occupied by anatomy, rather than the result of broad variation across the brain such as a gradient. Further, this transcriptional variation is adequately described by linear combinations of individual gene weights. This would fit with biology where a few localized transcriptional programs determine tissue function on a background of commonly expressed genes.

The detailed extrinsic ground truth of anatomy made AMBA an ideal dataset for this comparison, but during analysis we found inherent properties of the data that could be used for optimization when anatomy is not available. Dice coefficient, or goodness of fit to ground truth, correlated with connected components score, and the number of unique features. These could themselves be a basis for learning in unlabeled datasets where assumptions can be made about the spatial properties of the biology being represented.

Previous use of DLSC reported low sparsity as optimal for anatomical representations by global analysis of fit to ground truth as measured by AMI, but deeper analysis showed a different relationship with the sparsity parameter. Low sparsity derives redundant, broadly active, and overlapping features that only represent anatomy after secondary clustering. Increasing sparsity derives more accurate representations of specific elements of anatomy, directly resolving more unique regions. DLSC was the least accurate method, but at optimal sparsity, generated nearly twice the number of unique representations.

By our metrics, SFt representations were more accurate than DLSC or SPCA even after parameter tuning, for the same number of features. This is likely due to how each method applies sparsity to feature detection. The only parameter for SFt beyond the input data is the number of representations to optimize over and is common to all methods tested [25]. The simple implementation of SFt, makes it an attractive method for future applications, but raises the question if this means of applying sparsity is truly self optimizing or just particularly compatible with this dataset. In future implementations SFt could be tuned for further sparsity by using a smoothed fractional norm in place of the $l_1$ norm [26].

Sparse representation learning methods were unique in not requiring secondary clustering to represent anatomy, making the weight matrices that comprise each feature intelligible. Representations were not simply correlatively expressed genes. Instead, genes that defined borders of anatomy from the inside or outside were both highly weighted. Within the hippocampus, Dock10, an established marker of CA2 was highly weighted in representations of every anatomical region that bordered it, despite all methods being blind to spatial information (CA1,

CA3, DG, CP) (**Table 1**) [27]. This example demonstrates how feature detection is different from differential expression and should not be used as an analog for expression analysis when dissecting the possible biology of features derived without ground truth. Highly correlated genes within a feature can have informative relationships to genes expressed in other features that would lead to higher or lower weighting in the gene list. The stochastic initialization of sparse methods can also change weights or entire features, as they are built independently. Direct representations only covered a fraction of total identified brain regions with redundancy in a few elements of anatomy. We were able to define isolated elements of anatomy with relative ease, but this was limited to what was testable with defined marker genes. The limits of which brain regions that can be directly represented, and which large areas with multiple signatures should be excluded to improve analysis, would have to be determined and validated for more obscure elements of anatomy.

SPCA delivered the most sparse gene lists, but we found that SFt representations could be even further compressed without significant loss of fidelity. In this pilot study, after training the model on the 3k gene dataset we reduced the input data to 580 genes and still generated a better fit to anatomical ground truth than other methods using a full dataset. We could further minimize our gene lists for individual representations by applying supervised learning to the highest ranked elements to optimize their weighting. This two-step approach allowed for high accuracy representations with a list of elements that can be curated and optimized for downstream applications, such as limiting the list to proteins with commercially available antibodies. basis for deriving representations of biology with descriptive molecular signatures from a spatial transcriptomic dataset. This study benefited from having testable ground truth for biological validation, but the resulting weight matrix of each feature offers an unbiased transcriptional signature that is generalizable to other morphologies. The anatomy of the wild type (WT) C57BL/6J mouse is well described, but much less so for knockout or transgenic mouse strains or for related species of interest. AMBA scale data acquisition is unlikely to be repeated, but analysis with sparse learning has reduced the number genes needed to describe anatomy to a scale tractable by Multiplexed error-robust fluorescence in situ hybridization (MERFISH) or similar multiplexed spatial transcriptomic methods. What once took years of work could now be largely recapitulated in a single session [28].

One obvious improvement that could be made to any sparse methods on spatial datasets would be consideration of neighboring voxels for unsupervised learning. In this study, spatial aspects of the dataset were implicit, with representations appearing compact and contiguous only by virtue of their fit to ground truth. Finer representations could potentially be achieved by adding a spatial convolution step to sparse learning, or using spatial metrics as the basis for a loss function. For nonspatial datasets, other inherent biology such as gene ontology could serve a similar role to leverage this versatile tool for representation learning with preserved gene information.

## Methods

### Allen Mouse Brain Atlas (AMBA)

The Allen brain Atlas project (http://mouse.brain-map.org) has provided a comprehensive set of ~20,000 gene expression profiles with cellular resolution in the male, 56-day old C57BL/6J mouse brain [29]. Image data were collected using the in situ hybridization method in sagittally-oriented slices with 200 μm inter-slice resolution and 25 μm thickness. The expression patterns were replicated in the coronal sections for ~4,000 genes of high neurobiological interest [12]. The expression patterns were reconstructed in 3D and registered to a Nissl stain-based reference atlas (Allen Reference Atlas; ARA). The expression of a gene within 200 μm

isotropic voxel is the average intensity of pixels in the pre-processed image called smoothed expression energy. Each 3D gene expression data consists of $67 \times 41 \times 58$ (rostral-caudal, dorsal-ventral, left-right) spatially-matched volumes [29].

## Data pre-processing

We followed the quality control measure implemented by [13]. The Pearson correlation coefficient was measured for each gene in coronal and sagittal experiments to find the higher-consistency dataset. Then 25% of genes with the lowest correlations were removed, resulting in a selection of 3,041 genes in coronal slices from which 2,941 still exist in the current version of the AMBA dataset, which we adopted for this study. The genes are selected from the experiments done on coronal slices because of the higher fidelity.

We extracted and concatenated the expression energy values in the brain for all dimensionality reduction methods to form a large 63,113 voxel $\times$ 2,941 gene matrix, $E_{(v,g)}$, where $v$ and $g$ denote voxels and genes. We then z-transformed the data for each gene so that the mean of each gene's expression across the voxels was zero, and the standard deviation was one.

## Representation learning methods

For PCA, KPCA, ICA, SPCA, and DLSC we used their implementation in the publicly available SciKit-Learn package in Python [30]). We selected 100 for the number of components in all methods. In the sparse methods, SPCA and DLSC, the amount of sparseness is adjustable by the coefficient of the $l_1$ penalty, given by the parameter $\alpha$, for which we tested different values of $\alpha = \{0.1, 1, 10, 20\}$. The kernels used in KPCA were the quadratic and cubic polynomial, rbf, and sigmoid functions.

## Sparse Filtering (SFt)

Sparse filtering is an unsupervised feature learning method that efficiently scales to handle large input dimensions. The single hyperparameter to tune is the number of features. SFt optimizes for population sparsity (a few non-zero features represent each sample), lifetime sparsity (each feature is active for a few samples), and high dispersal (features share similar statistics) [31].

The math proposed for SFt [20] considers the feature distribution matrix indexed by j for features as rows and by i for samples as columns.

In the corresponding feature distribution $f_j^{(i)}$ represents the $j^{th}$ feature value (rows) for the $i^{th}$ example (columns), resulting in $f_j^{(i)} = w_j^T x^{(i)}$. The first step is to normalize each feature by dividing it by its $l_2$ norm across all examples $\tilde{f}_j = f_j / ||f_j||_2$. The features are then $l_2$ normalized across examples $\hat{f}^{(i)} = \tilde{f}^{(i)} / ||\tilde{f}^{(i)}||_2$. Then the $l_1$ penalty is used to optimize the features for sparsity so that in a dataset of M examples, the objective becomes:

$$minimize \ \sum_{i=1}^{M} ||\hat{f}^{(i)}||_1 = \sum_{i=1}^{M} ||\frac{\tilde{f}^{(i)}}{||\tilde{f}^{(i)}||_2}||_1.$$

We used the open-source GitHub repository of Sparse Filtering in Python (https://github.com/jmetzen/sparse-filtering) in this study. This software package transposes the voxel $\times$ gene matrix, $E_{(v,g)}$, to be consistent with the Matlab code provided by (Ngiam et al., 2011). It also incorporated the soft-absolute activation function and Limited-memory Broyden, Fletcher, Goldfarb, and Shanno (L-BFGS) minimizer.

## K-means clustering

After data representation using previously mentioned methods, we applied the unsupervised K-means clustering method. We used the publicly available K-means clustering

implementation in SciKit-Learn (Pedregosa et al., 2011), and the number of clusters (K) ranged from 1 to 50 with the step of 1, and 50 to 550 with the step of 50.

## Similarity measures

We chose Adjusted Mutual Information (AMI) and Adjusted Rand Index (ARI) in the SciKit-Learn package [30] to measure the similarity of clustered representations with anatomy, where the adjusted versions of MI and RI account for the chance.

We developed a function to compute the DICE similarity coefficient to measure the spatial overlap of brain regions and representations (code example available on GitHub). It ranges from 0, for no overlap, to 1, showing a perfect overlap. The formula to calculate DSC for two regions (A and B) is:

$$DSC_{(A,B)} = 2(A \cap B)/(A + B)$$

where $\cap$ is the intersection [32].

## Logistic regression

For Sparse Filtering, we found the candidate pairs of brain regions and features by measuring the Dice coefficient of a given brain region against a thresholded feature. We thresholded the features using K-means clustering, with the single feature of interest as input, and choosing the number of clusters at two. We compared both labels against the region of interest using the Dice coefficient, and kept the better of the two. We repeated this procedure for every brain region against all features, and kept the best region-feature combination.

We chose the brain regions CA1, CA3, and CP for logistic regressions, as they were among the top overlapping combinations for SFt and other methods. We determined the gene inputs for logistic regressions by identifying the largest ten weights by absolute value associated with a given feature produced by a transformation, and then finding the genes associated with these weights (**Table 1**). We then mean-centered and standard-scaled these ten genes, and prepared a shuffled five-fold cross-validation set. We trained a logistic regression from the normalized genes onto the region of interest, and reported the ROC-AUC score on the test set. Normalization, logistic regression, shuffling, cross-validation, and ROC-AUC score were all performed using their respective implementations in the scikit-learn library [30].

## Sparsity metrics

Here we described the metrics used to measure sparsity and provided the code examples on GitHub.

## Feature sparsity

To measure the feature sparsity, first, we found the number of active voxels in each feature by measuring the mean value of voxels in each feature and counting the number of voxels with values above that. Then we used the ratio below to measure sparsity:

Feature Sparsity = Total number of voxels in the feature / Number of active voxels

## Weight sparsity

To measure the weight (gene) sparsity, first, we found the number of active genes in each feature transformation by measuring the mean value of genes in each feature and counting the number of genes with values above that. Then we used the ratio below to measure sparsity:

Weight (gene) Sparsity = Total number of genes / Number of active genes in the feature

## Connected components

We determined the number of connected components within a feature using the implementation of marching cubes in the scikit-image library [30] to develop an adjacency matrix, and an algorithm that determines the number of connected components in a graph from its adjacency matrix available in the NetworkX library [33].

We reconstructed the representations generated by each method into the anatomical space, then applied the scikit-image implementation of marching cubes [34] to the resulting 3D array, using the mean value of the feature as the threshold level for the method. The output of this method is a set of vertices and faces that comprise a set of 3D surfaces. We constructed an adjacency matrix for the set of vertices using the set of faces, and found the number of connected components by passing this adjacency matrix to the number_connected_components algorithm in the NetworkX library [33].

## Shannon entropy

We measured the Shannon entropy of a feature by binning out the continuous values of the feature into discrete levels, and treated each discrete value as though it were a unique symbol in the Shannon entropy formula:

$$H(x) = -\sum_{i=1}^{n} P(x_i) \, ln \, P(x_i)$$

where each $x_i$ in $i = [1, n]$ is a discrete value of the binned feature.

## Spatial entropy

We developed a measurement of the spatial entropy of a feature using an extension of the gray level co-occurrence matrix method in use on 2D images. Pixel intensities present in an image are binned into discrete levels, and instances of co-occurrence of these levels is counted with specified spatial relationships as the elements of a matrix. We extended this method to a 3D image by including a three-dimensional relationship [35]. Spatial entropy of an image can be measured by finding the Shannon entropy of this matrix, in which each (i,j) index of the matrix is as though it were a unique symbol in the Shannon entropy formula described previously.

## Acknowledgments

Thanks to Dr. Chris Plaisier for consultation on our metrics, Jan Hendrik Metzen for sharing his implementation of SFt, and Research Computing at Arizona State University for providing the high-performance computing and data storage used to generate the analyses reported within this paper.

## Author Contributions

**Conceptualization:** Mohammad Abbasi, Connor R. Sanderford, Mirjeta Pasha, Benjamin B. Bartelle.

**Formal analysis:** Mohammad Abbasi, Connor R. Sanderford.

**Funding acquisition:** Benjamin B. Bartelle.

**Investigation:** Mohammad Abbasi.

**Methodology:** Mohammad Abbasi, Connor R. Sanderford.

**Project administration:** Benjamin B. Bartelle.

**Resources:** Benjamin B. Bartelle.

**Software:** Connor R. Sanderford, Narendiran Raghu.

**Supervision:** Mirjeta Pasha, Benjamin B. Bartelle.

**Visualization:** Connor R. Sanderford, Narendiran Raghu.

**Writing – original draft:** Benjamin B. Bartelle.

**Writing – review & editing:** Mohammad Abbasi, Connor R. Sanderford, Benjamin B. Bartelle.

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
