## [Decision Letter · Decision Letter 0]

22 Jul 2022

PONE-D-22-12274Sparse Representation Learning Derives Biological Features with Explicit Gene Weights from the Allen Mouse Brain AtlasPLOS ONE

Dear Dr. Bartelle,

Thank you for submitting your manuscript to PLOS ONE. After careful consideration, we feel that it has merit but does not fully meet PLOS ONE’s publication criteria as it currently stands. Therefore, we invite you to submit a revised version of the manuscript that addresses the points raised during the review process.

Please, revise the manuscript to accentuate the goals and conclusions/achievements.  Please, incorporate biological interpretation of the results. The method section could be revised to give a brief description of all the methods and their associated hyperparameters (with tested ranges). A lot of observations in the paper are simply reported and a follow-up discussion about their potential reasons and implications could improve the narrative.

Please, address technical issues raised by the reviewers:

 Visual clustered representations in the area of the olfactory bulb and nucleus will be less accurate in these regions due to registration issues. The brains were distorted somewhat in these areas through processing. Please, clarify what do you mean by a single coronal slice as referred to on line 144.

Application of these methods to sagittal oriented sections may be problematic due to the limited resolution in this orientation.

 What precisely is the difference in results of SVD versus PCA methods here, as the former is essentially equivalent at full rank. What anatomical features differ?

 Please, consider providing a basic explanation of how the application of sparse representation learning algorithms work in this application.

 References to advances in applied information theory should be given.

 Long sections titles such as “Variance based…”

on line 151 are overly descriptive without the readers ability to parse meaning. A more intuitive section title with explaining the result early in the first paragraph would be a little easier on the reader.

Have unsupervised representation learning methods never been applied to or studied for the AMBA dataset? If so, why is that the case and how does this paper address it?

It seems the hyperparameters for different algorithms have been selected using the performance on ground truth labels. How will these be selected for other datasets? One experiment could be to split the existing dataset into different sets and see if the hyperparameters selected for one set generalize to another. It is important to answer how can SFt (the proposed method) generalize to other similar datasets and be useful for the community?

How do the applied methods in the paper help fix the drawback “Initial dimensionality reduction filters for high variance global trends over localized features”?

“ AMI scores peaked well below the 574 labeled brain regions, with most fitting optimally at ~200 clusters” - what does it imply for the applicability of these methods in the domain?

How is the feature selection done using the sparse methods affected by the correlation structures in the data? Would these methods suffer from identifiability issues due to highly correlated genes?

For data compression, how was the threshold selected to filter out the genes at each step? Why was the SFt step run multiple times? Was the original gene list analyzed with different thresholding?

What is the significance of the other genes reported in the list of 12 highest weighted genes that are not associated with the brain region (highlighted in orange)? For example, what is the role of Pip5k1a that appears for 2 different regions?

Table 1 could be better formatted and maybe have the relevant genes in bold text.

Some figures have empty white spaces on the sides.

Does the logistic regression task have any class label imbalance? If so, AUPRC score might be worth observing as well.

We look forward to receiving your revised manuscript.

Kind regards,

Gennady S. Cymbalyuk, Ph.D.

Academic Editor

PLOS ONE

Journal Requirements:

Reviewers' comments:

Reviewer's Responses to Questions

**Comments to the Author**

1. Is the manuscript technically sound, and do the data support the conclusions?

Reviewer #1: Yes

Reviewer #2: Partly

2. Has the statistical analysis been performed appropriately and rigorously? 

Reviewer #1: Yes

Reviewer #2: No

3. Have the authors made all data underlying the findings in their manuscript fully available?

Reviewer #1: Yes

Reviewer #2: Yes

4. Is the manuscript presented in an intelligible fashion and written in standard English?

Reviewer #1: Yes

Reviewer #2: Yes

5. Review Comments to the Author

Reviewer #1: Sparse Representation Learning Derives Biological Features with Explicit Gene Weights from the Allen Mouse Bran Atlas is an interesting and comprehensive view of the relationship between spatial transcriptomic patterns and neuroanatomy. There have been several papers over the years since the release of the Allen Mouse Brain Atlas, but this study is among the most careful and detailed. The authors have a solid understanding of the issues of transcriptomic defined clusters and marker gene identification. To eliminate the need for dimension reduction and clustering the authors explored sparse learning methods for building localized features from minimal numbers of inputs. This topic is also highly relevant to the more modern multiplex spatial transcriptomics data sets and the authors may wish to comment on this point.

The authors use the 200micron isotropic voxel-based data from the Allen Atlas as a basis for studying the detailed correspondence of genetics with neuroanatomy over 574 anatomically labelled voxels. The methods studied here provide a natural means of identifying minimal marker sets of genes defining anatomic patterning. The range of k-means clusters used is appropriate and covers up to the right order of the number of voxels. The use of these methods to determine an optimal fitting at around 200 clusters and using sparse methods is an interesting result and may indicate a fundamental level of resolution in the data. Finally, the paper is written rather technically from an informatics perspective for this journal, and I might recommend shortening the work somewhat with a it more biological interpretation for readers.

Major

• Visual clustered representations in the area of the olfactory bulb and nucleus will be less accurate in these regions due to registration issues. The brains were distorted somewhat in these areas through processing. I’m not certain what the authors mean by a single coronal slice as referred to on line 144.

• Application of these methods to sagittal oriented sections may be problematic due to the limited resolution in this orientation. The results of the paper are ample with restriction to the coronal datasets in this reviewer’s opinion.

• What precisely is the difference in results of SVD versus PCA methods here, as the former is essentially equivalent at full rank. What anatomical features differ?

• The lack of improvement of KPCA to the parcellations is interesting. What is the authors intuition for this result?

• The application of sparse representation learning algorithms is interesting and important I this context. It might be helpful for the authors to give a basic explanation of how these methods work in this context for the reader.

• The use of this approach to identify more minimal sets of defining gene sets is an important result and highlight of the work. I would make more of this section and produce a comprehensive defining set by anatomic region to the extent possible, together with biological annotations where available.

• One important consideration for this work which I think should be remarked discussion and presented as a caveat is that the anatomic labels are determined by neuroanatomists interpretation of the definition of these regions. This itself has variability and potential discrepancy from what might be considered ground truth, whatever that might be. Thus, we are making a comparison with annotations themselves which are potentially inaccurate. This in no way diminishes the importance of the approach but should be remarked. With respect to the last point, a comprehensive set of tables showing how predicted transcriptomic regions using the various approaches intersects with the Allen Reference Atlas anatomy would be useful. A abbreviated form of this could be given in the main paper and the full set supplementally.

Minor

• References to advances in applied information theory should be given.

• Long sections titles such as “Variance based…”

on line 151 are overly descriptive without the readers ability to parse meaning. A more intuitive section title with explaining the result early in the first paragraph would be a little easier on the reader. These types of titles are attractive but if they become too long it starts to mix with the actual description in the paper.

Reviewer #2: This paper applies and compares various unsupervised representation learning methods - Independent Component Analysis (ICA), Principal Component Analysis (PCA), Kernel PCA (KPCA), Sparse PCA (SPCA), Dictionary Learning and Sparse Components (DLSC), and Sparse Filtering (SFt) - to the transcriptomic data from the Allen Mouse Brain Atlas (AMBA) project. Given the ground truth anatomy labels, the paper evaluates the quality of the representation learned by different algorithms and ways to generate gene lists and compressed information. The paper first demonstrates that out of the applied unsupervised methods (+ K-means to assign anatomical label) SFt gives the best AMI and ARI scores that test if the cluster labels match the ground truth labels. The paper then performs other analyses and makes multiple observations like - (1) SFt overall gives good performance for a variety of evaluation metrics like DICE scores etc. (2) Sparse representation learning methods without secondary clustering provide a ranked gene list of samples (3) The ranked list can be used to perform data compression and accurate supervised classification of regions.

While the paper presents potentially useful results, the main contribution of the work is unclear. It would be very useful for the paper to have a single coherent message of how its observations could help the community assess/better analyze the new spatial transcriptomics datasets. Specifically, addressing the following questions/suggestions:

Major comments:

Have unsupervised representation learning methods never been applied to or studied for the AMBA dataset? If so, why is that the case and how does this paper address it?

The paper focuses on only one dataset, making it hard to assess the robustness of these methods. All of the results and conclusions were dependent on the availability of ground truth data. Can the paper include some other datasets to see if similar conclusions can be drawn if the ground truth is hidden?

For example, it seems the hyperparameters for different algorithms have been selected using the performance on ground truth labels. How will these be selected for other datasets? One experiment could be to split the existing dataset into different sets and see if the hyperparameters selected for one set generalize to another. It is important to answer how can SFt (the proposed method) generalize to other similar datasets and be useful for the community?

How do the applied methods in the paper help fix the drawback “Initial dimensionality reduction filters for high variance global trends over localized features”? I am assuming this was the reason to exclude tSNE and other methods mentioned earlier in the comparison.

“ AMI scores peaked well below the 574 labeled brain regions, with most fitting optimally at ~200 clusters” - what does it imply for the applicability of these methods in the domain?

How is the feature selection done using the sparse methods affected by the correlation structures in the data? Would these methods suffer from identifiability issues due to highly correlated genes?

For data compression, how was the threshold selected to filter out the genes at each step? Why was the SFt step run multiple times? Was the original gene list analyzed with different thresholding?

What is the significance of the other genes reported in the list of 12 highest weighted genes that are not associated with the brain region (highlighted in orange)? For example, what is the role of Pip5k1a that appears for 2 different regions?

Minor comments:

The paper could be revised to highlight the main conclusions and contributions of the work.

The method section could be revised to give a brief description of all the methods and their associated hyperparameters (with tested ranges).

Table 1 could be better formatted and maybe have the relevant genes in bold text.

Some figures have empty white spaces on the sides.

A lot of observations in the paper are simply reported and a follow-up discussion about their potential reasons and implications could improve the narrative.

Does the logistic regression task have any class label imbalance? If so, AUPRC score might be worth observing as well.

6. PLOS authors have the option to publish the peer review history of their article (what does this mean?). If published, this will include your full peer review and any attached files.

Reviewer #1: **Yes: **Michael Hawrylycz

Reviewer #2: No

---

## [Author Response · Author response to Decision Letter 0]

31 Aug 2022

All responses are included in the attachment labeled "Response to reviewers."

We thank the reviewers for their insightful comments. We have taken each into account for this manuscript and it has given us food for thought as we continue developing sparse learning methods on AMBA and other datasets. Sections of the text marked in blue denote additions to the manuscript, including methods for new suggested analyses. 

We have expanded the manuscript to provide more background and edited where requested to improve the narrative flow. Specifically, we have added a new section to the results to more gradually introduce the concept of interpreting learned weight matrices as signature gene lists, and 4 new paragraphs to the discussion section to better interpret the results and point to future applications. We have also expanded descriptions of the learning tools used in our introduction and methods section to make the methods more accessible to a broader audience.

To address specific comments, please see below.

 Visual clustered representations in the area of the olfactory bulb and nucleus will be less accurate in these regions due to registration issues. The brains were distorted somewhat in these areas through processing. Please, clarify what do you mean by a single coronal slice as referred to on line 144.

You will find a clarification between pg 7 and 8. It was important to note these artifacts as they were similar to those discussed in Ortiz 2020, where they were manually curated from their implementation of ICA.

Application of these methods to sagittal oriented sections may be problematic due to the limited resolution in this orientation.

We have included this caveat in the same section and note that our analysis was performed on the volumetric data of AMBA. While the reported resolution of the dataset is 200µm isotropic, the data quality mismatch between sagittal and coronal collected data is likely the root of most artifacts. Much of the data was intractable outside of the set originally curated by Ng and colleagues. 

 What precisely is the difference in results of SVD versus PCA methods here, as the former is essentially equivalent at full rank. What anatomical features differ?

On page 9 we clarify our use of PCA as equivalent to SVD and that we used the same number of components to recapitulate the Bohland 2010 analysis. We use the term PCA only because it is more familiar to a broad audience.

 Please, consider providing a basic explanation of how the application of sparse representation learning algorithms work in this application. References to advances in applied information theory should be given.

On page 11 we provide this explanation and relate our findings back to the major points of this introduction. Several references have been added including text book references to ridge (l2) and LASSO (L1) regression.

 Long sections titles such as “Variance based…”on line 151 are overly descriptive without the readers ability to parse meaning. A more intuitive section title with explaining the result early in the first paragraph would be a little easier on the reader.

We have changed the section headings to make them more accessible to a general audience.

Have unsupervised representation learning methods never been applied to or studied for the AMBA dataset? If so, why is that the case and how does this paper address it?

On page 16 we introduce a previous analysis using sparse learning, compare it to other methods, and demonstrate how the reported optimal parameters, while correct for their published analysis, present a surprising artifact in that 95% of the derived features represented the same element of anatomy. This result provides a useful insight into how to best implement sparse learning and contributes to the value of the paper.

It seems the hyperparameters for different algorithms have been selected using the performance on ground truth labels. How will these be selected for other datasets? One experiment could be to split the existing dataset into different sets and see if the hyperparameters selected for one set generalize to another. It is important to answer how can SFt (the proposed method) generalize to other similar datasets and be useful for the community?

On page 26 we discuss how the compressed gene list derived from this analysis could be applied to biology that lacks ground truth labels. We achieved compression by eliminating low rank genes and repeating SFt, essentially splitting the dataset in an informed way.

 How do the applied methods in the paper help fix the drawback “Initial dimensionality reduction filters for high variance global trends over localized features”?

On page 4 and 11 we cite how constraining our learning for sparse signatures differs from variance based methods.

“ AMI scores peaked well below the 574 labeled brain regions, with most fitting optimally at ~200 clusters” - what does it imply for the applicability of these methods in the domain?

On page 6 and 25 we discuss how resolution limits identification of many smaller brain regions and how the strong signatures can occlude more subtle features. Our major point stands in that learning methods are necessary for complex datasets and for transcriptomic data at least, sparsity constraints return a better fit to ground truth.

How is the feature selection done using the sparse methods affected by the correlation structures in the data? Would these methods suffer from identifiability issues due to highly correlated genes?

On page 25 we further discuss this intriguing question. In our experience gene weights can change drastically across rounds of learning even without changing the data. Our approach to compression was based on this observation where genes were systematically eliminated, then learning was performed. 

For data compression, how was the threshold selected to filter out the genes at each step? Why was the SFt step run multiple times? Was the original gene list analyzed with different thresholding?

On page 19 and 20 we have expanded our results section to introduce this idea better and explain the approach. Without repeating learning, removed values would be considered as real 0 values in the previous model. It was necessary to build a new learning model at each step to generate a new matrix. Using the same number of clusters was also necessary for a valid comparison across steps.

There are several optimizations suggested by these results that we are now following up on to develop better informed approaches to gene compression. Those results are necessarily application focused and beyond the scope of this paper. 

What is the significance of the other genes reported in the list of 12 highest weighted genes that are not associated with the brain region (highlighted in orange)? For example, what is the role of Pip5k1a that appears for 2 different regions?

On page 25 we discuss how expression levels within a feature can differ from informativeness of expression and present Dock10 as an example. This gene is a marker of CA2, but is highly weighted in each presented representation. Marker genes in general were highly weighted, even when their expression was adjacent to the learned features. We explicitly state feature learning is not a substitute for differential expression analysis.

Table 1 could be better formatted and maybe have the relevant genes in bold text.

We have bolded marker genes for their respective anatomy. 

Some figures have empty white spaces on the sides.

We have reformatted figures and legends for this new draft.

Does the logistic regression task have any class label imbalance? If so, AUPRC score might be worth observing as well.

On page 21 we have included AUPRC scores, which also perform well. We have included their generation in the methods section on page 31.

We attempted to address all other comments to the best of our ability and we once again thank the reviewers for their time. We appreciate the opportunity to make this the best manuscript it can be and we are excited to build on our findings.

Thank you.

Benjamin Bartelle PhD

Arizona State University

---

## [Decision Letter · Decision Letter 1]

9 Feb 2023

Sparse Representation Learning Derives Biological Features with Explicit Gene Weights from the Allen Mouse Brain Atlas

PONE-D-22-12274R1

Dear Dr. Bartelle,

We’re pleased to inform you that your manuscript has been judged scientifically suitable for publication and will be formally accepted for publication once it meets all outstanding technical requirements.

Kind regards,

Gennady S. Cymbalyuk, Ph.D.

Academic Editor

PLOS ONE

Additional Editor Comments (optional):

Reviewers' comments:

Reviewer's Responses to Questions

**Comments to the Author**

1. If the authors have adequately addressed your comments raised in a previous round of review and you feel that this manuscript is now acceptable for publication, you may indicate that here to bypass the “Comments to the Author” section, enter your conflict of interest statement in the “Confidential to Editor” section, and submit your "Accept" recommendation.

Reviewer #1: All comments have been addressed

Reviewer #3: (No Response)

2. Is the manuscript technically sound, and do the data support the conclusions?

Reviewer #1: Yes

Reviewer #3: Partly

3. Has the statistical analysis been performed appropriately and rigorously? 

Reviewer #1: Yes

Reviewer #3: N/A

4. Have the authors made all data underlying the findings in their manuscript fully available?

Reviewer #1: Yes

Reviewer #3: Yes

5. Is the manuscript presented in an intelligible fashion and written in standard English?

Reviewer #1: Yes

Reviewer #3: Yes

6. Review Comments to the Author

Reviewer #1: The authors have addressed all of my questions and concerns and provided much additional information related to other reviewers critiques.

Reviewer #3: This is a would-be interesting paper studying contributions of individual genes to some feature using unsupervised learning methods, using the spatial transcriptomic data and anatomical labels of the Allen Mouse Brain Atlas as a test dataset.

In introduction authors mentioned that some other methods have weaknesses, and they suggested that used by authors methods are better. I would not look at these methods as competitors. They are just different tools; it is better to take advantage of each of them rather than using just one that currently give best solution.

For example, to find what genes are representing major features it will be better use methods that less sensitive to rare genes. But additionally, one could study how combination of rare or low expressed genes could be related to other features.

It would be interesting to compare Table 1 with results of other methods.

pp.4-5. Authors wrote: “Initial dimensionality reduction filters for high variance global trends over localized features, offering low sensitivity to rare or low expressing genes (Torgerson, 1952).”

This is not weakness of method but rather weakness of dataset, as authors approach offering low sensitivity to less represented features in used dataset.

Authors wrote: “the contribution of any one gene to a cluster is not explicit”. Actually, suggested by authors approach as any other approach based on statistics is not explicit as well.

Methods section should be before results and discussion.

Unreadable characters at page 16, Figure 4, pages 39-41, 45

7. PLOS authors have the option to publish the peer review history of their article (what does this mean?). If published, this will include your full peer review and any attached files.

Reviewer #1: **Yes: **Michael Hawrylycz

Reviewer #3: No

---

## [Editor Report · Acceptance letter]

20 Feb 2023

PONE-D-22-12274R1 

Sparse Representation Learning Derives Biological Features with Explicit Gene Weights from the Allen Mouse Brain Atlas 

Dear Dr. Bartelle:

I'm pleased to inform you that your manuscript has been deemed suitable for publication in PLOS ONE. Congratulations! Your manuscript is now with our production department. 

Kind regards, 

on behalf of

Dr. Gennady S. Cymbalyuk 

Academic Editor

PLOS ONE